# Decreased motor cortex excitability mirrors own hand disembodiment during the rubber hand illusion

Francesco della Gatta[1,2†], Francesca Garbarini[3*†], Guglielmo Puglisi[2], Antonella Leonetti[2], Annamaria Berti[4], Paola Borroni[2]

[1]Department of Philosophy, University of Milan, Milan, Italy; [2]Department of Health Sciences, University of Milan Medical School, Milan, Italy; [3]SAMBA Research Group, Department of Psychology, University of Turin, Turin, Italy; [4]Neuroscience Institute of Turin, University of Turin, Turin, Italy

**Abstract** During the rubber hand illusion (RHI), subjects experience an artificial hand as part of their own body, while the real hand is subject to a sort of 'disembodiment'. Can this altered belief about the body also affect physiological mechanisms involved in body-ownership, such as motor control? Here we ask whether the excitability of the motor pathways to the real (disembodied) hand are affected by the illusion. Our results show that the amplitude of the motor-evoked potentials recorded from the real hand is significantly reduced, with respect to baseline, when subjects in the synchronous (but not in the asynchronous) condition experience the fake hand as their own. This finding contributes to the theoretical understanding of the relationship between body-ownership and motor system, and provides the first physiological evidence that a significant drop in motor excitability in M1 hand circuits accompanies the disembodiment of the real hand during the RHI experience.

**\*For correspondence:** francesca. garbarini@unito.it

[†]These authors contributed equally to this work

**Competing interests:** The authors declare that no competing interests exist.

## Introduction

The sense of body ownership (i.e. the belief that a specific body part belongs to one's own body) (*Gallagher, 2000*) is a fundamental aspect of self-consciousness. Apparently, in normal conditions, the feeling of body ownership does not need any particular explanation; it is immediate and even obvious. However, both pathological cases after brain damage (somatoparaphrenia [*Romano et al., 2014*] and pathological embodiment [*Pia et al., 2016*; *Fossataro et al., 2016*; *Garbarini et al., 2013*, *2014*, *2015*; *Pia et al., 2013*; *Garbarini and Pia, 2013*]) and experimental manipulations in healthy subjects (e.g. the rubber hand illusion – RHI [*Botvinick and Cohen, 1998*]) suggest that body ownership, as we normally experience it, is the product of many different and complex operations. It has been suggested that the feeling that our body belongs to us presumably depends on multisensory integration processes arising within a fronto-parietal network, where sensory inputs coming from different modalities are realigned in a unique reference frame (*Blanke et al., 2015*). Within this network, the ventral premotor cortex seems to play a crucial role, thus establishing, both in monkeys (*Graziano, 1999*) and in humans (*Makin et al., 2008*; *Ehrsson et al., 2004*), an anatomical link between the sense of body ownership and the motor system. Furthermore, it has been proposed that voluntary motor activity of body parts contributes critically to the subjective experience of body ownership (*Tsakiris et al., 2010*). Within this context, we asked whether the subjective and sometimes illusory sense of body ownership influences objective measures of the sensory-motor system. We took advantage of the RHI paradigm in order to provide a physiological counterpart of the interaction between body awareness and motor control, investigating the relationship between

**eLife digest** The feeling of body ownership — that the various parts of your body are all part of you — is something that we typically take for granted. However, brain damage can disrupt this sensation and leave individuals convinced that an arm or a leg is no longer their own. Even in healthy people, the 'rubber hand illusion' can temporarily produce a similar phenomenon. Individuals watch a lifelike rubber hand being touched while their own hand – which is hidden from view – is touched at the same time. This creates the feeling that the artificial hand has become part of their body, while their real hand is left in a 'disembodied' state.

How does the brain generate this illusory sense of ownership and accompanying sense of disembodiment? A person's ability to move their body is thought to contribute to their feeling of body ownership. Therefore, della Gatta, Garbarini et al. asked whether the brain's ability to move the real hand changes during the rubber hand illusion.

In the experiments, the region of the brain that controls hand movement was artificially stimulated in a number of volunteers. When an individual had been primed by the rubber hand illusion to perceive a fake hand as part of their own body, their brain was temporarily less able to activate the muscles of their real hand. This is as if the brain no longer considered the real hand as part of the body. Thus, the altered sense of body ownership experienced during the rubber hand illusion is not a bizarre fantasy, but corresponds to a physiological reaction that accompanies changes in brain activity.

The next step is to further define and quantify the relationship between the sense of body ownership and the control of body movement. Specifically, how does activity in the brain areas that control movement contribute to the sense of body ownership? And how do these brain regions communicate with one another to generate a sense of self?

body ownership alterations, such as those occurring during the RHI, and modulation of primary motor cortex excitability.

During the RHI, the subject's real hand is out of view, while a realistic rubber hand (RH) is positioned in its place. When the experimenter synchronously strokes the index finger of both the real and the fake hand, most subjects, after a few seconds of viewing the fake hand's finger being touched, attribute their tactile sensation to the RH hand, which they start to perceive as their own. During the illusion, the subject's hand-centered reference frame shifts towards the RH, and so it has been proposed that, as a consequence, the real hand is subject to a sort of disembodiment (*Ehrsson et al., 2004*). Accordingly, a feeling of disownership of one's own hand has been reported as an important behavioural component of the RHI (*Longo et al., 2008*), while a decrease in the temperature of the real (disembodied) hand has been observed as a neurophysiological correlate of the RHI (*Moseley et al., 2008*). Moreover, it has been demonstrated that cooling the subject's hand increases the strength of the RHI, whereas warming the hand decreases it (*Kammers et al., 2011*). However, another study found that hand-cooling can be present in both the experimental (synchronous) and the control (asynchronous) condition, thus suggesting that it is not a reliable correlate of the subjective feeling of hand disownership in the RHI (*Rohde et al., 2013*). A further study proposed that somatosensory changes observed in the participants' hand while experiencing the RHI can be explained by cross-modal mismatch between the seen and felt position of the hand, and are not necessarily a signature of disownership (*Folegatti et al., 2009*).

In the present study, in the main behavioral experiment, we employed a classical RHI procedure to investigate the presence of the illusory experience in our subject sample. In addition, in the control behavioral experiment, the complementary presence of both embodiment of the RH and disembodiment of the real hand was explicitly investigated. Moreover, during the main physiological experiment, we studied the excitability modulation of motor circuits to the real (stimulated) hand during RHI. While subjects received visual-tactile stimulations, either synchronous (to induce the illusion) or asynchronous (control condition), motor evoked potentials (MEPs) were elicited by a single-pulse of transcranial magnetic stimulation (TMS) over the left primary motor cortex (M1) and

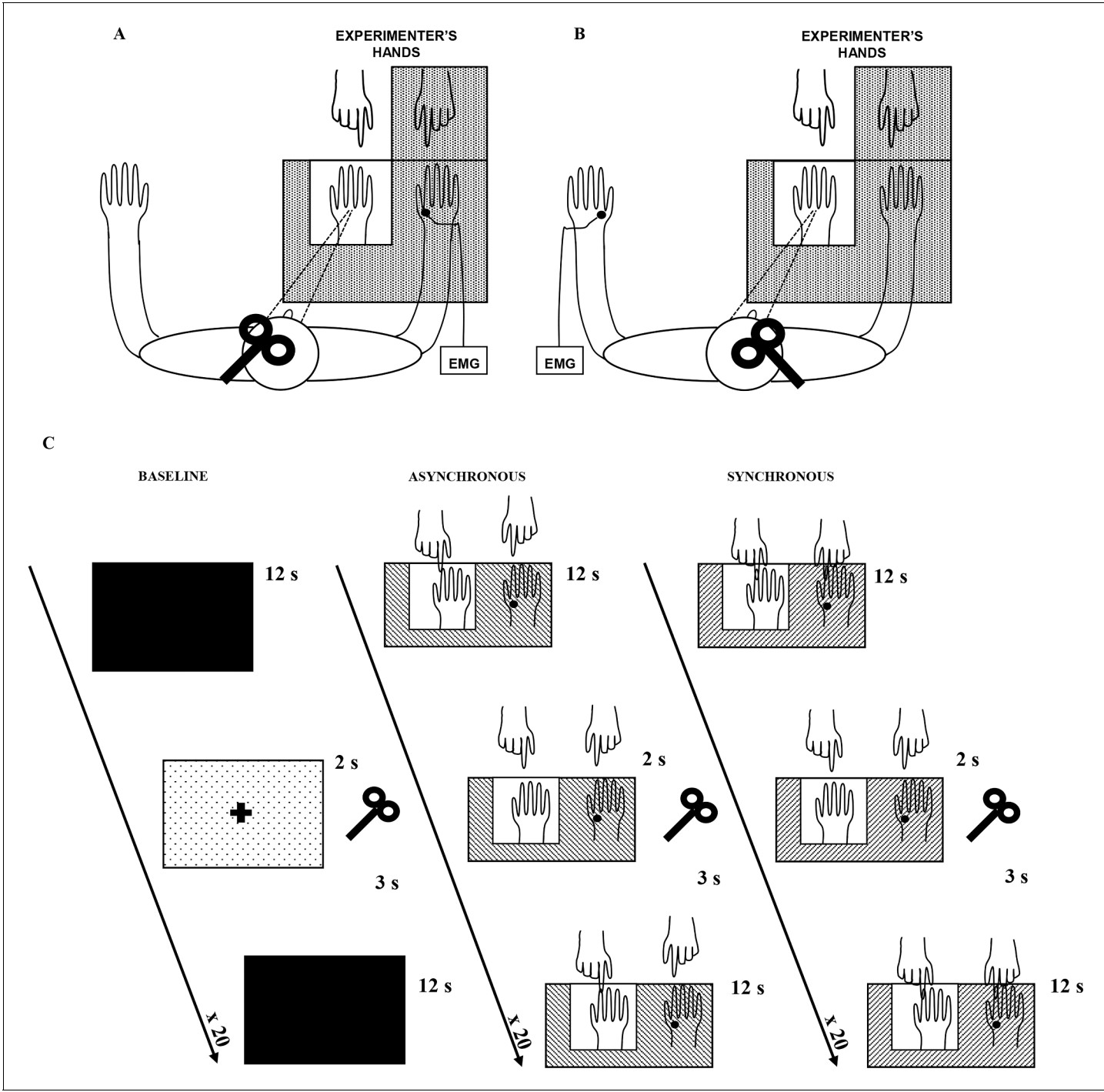

**Figure 1.** Experimental setup. The white square indicates the opening in the experimental wooden box through which the rubber hand is visible to the subject. Subjects could see only the rubber hand being stroked by the experimenter's right hand. In A, main experiment, MEPs were acquired from the stimulated (right) hand's FDI muscle; in B, control experiment, MEPs were acquired from non-stimulated (left) hand's FDI muscle. In C, timeline of the study and experimental conditions are plotted.

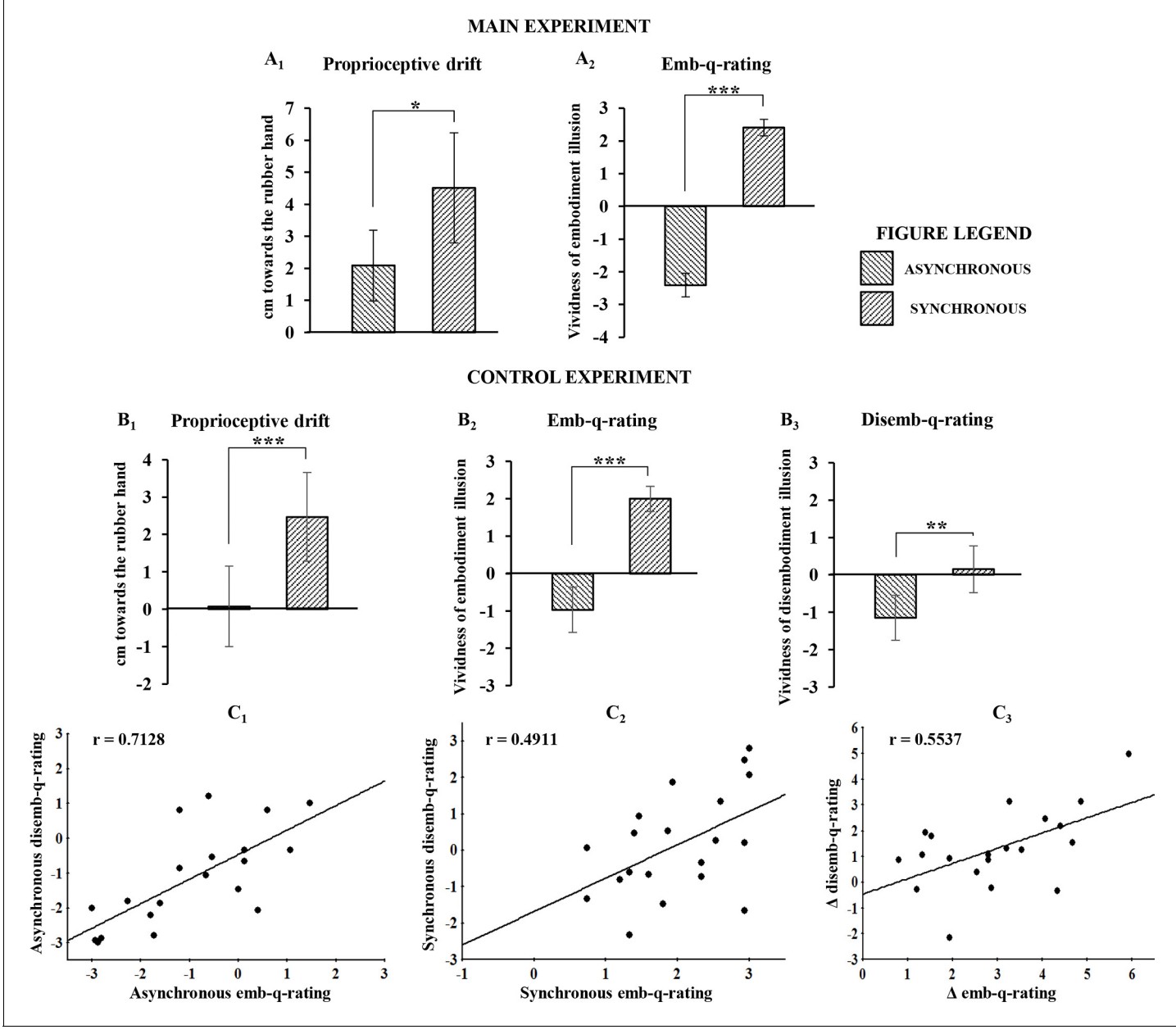

**Figure 2.** Behavioral results following asyncronous and syncronous condition. The average values for proprioceptive drift and emb-q-rating are plotted in $A_1$ and $A_2$, respectively, for the main experiment, and in $B_1$ and $B_2$, respectively, for the control experiments. In $B_3$ are reported average values for disemb-q-rating. Error bars indicate 95% CI. Significant levels: $*p<0.05$; $***p<0.0001$. Linear regressions between emb-q rating and disemb-q-rating in both syncronous and asyncronous conditions and in the delta syncronous minus asyncronous are plotted in $C_1$, $C_2$, $C_3$, respectively. All subjects behavioural data are available in the additional source data file (see *Figure 2—source data 1* and *2*).

The following source data is available for figure 2:

**Source data 1.** Main experiment behavioral results following asyncronous and synchronous condition.
**Source data 2.** Control experiment behavioral results following asyncronous and synchronous condition.

recorded from the right first dorsal interosseous muscle (FDI). See details in Material and methods and in *Figure 1A*.

We hypothesized that, in the motor domain, a disembodiment effect during the RHI might be measurable as a lower excitability of motor pathways to the real hand, i.e. a situation in which a stronger voluntary command is needed to bring enough motor neurons to threshold and thus to elicit movement. Thus, a decrease of FDI MEP amplitude compared to the baseline, specific to the real (disembodied) hand, was expected in the synchronous condition (where the subjects experienced the RHI) and not in the asynchronous (control) condition. Furthermore, according to behavioral studies reporting an increased illusory experience over time (*Lewis and Lloyd, 2010*; *Valenzuela Moguillansky et al., 2013*), this inhibitory motor response was expected to increase during exposure to the illusion. Finally, we expected the MEP amplitude decrease to be specific for the stimulated (right) hand, i.e. no amplitude modulation in the non-stimulated (left) hand (see *Figure 1B*; control physiological experiment).

## Results

The main behavioral results showed that both proprioceptive drift towards the RH and embodiment questionnaire rating (emb-q-rating) were significantly higher in the synchronous than in the asynchronous condition (drift=mean ± sd: 4.51 ± 4.2 cm vs. 2.08 ± 2.75 cm; $t_{(23)}$=2.783, p=0.0105, dz=0.58; emb-q-rating=mean ± sd: 2.4 ± 0.64 vs. −2 ± 0.9, Z=4.2857, p=0.000018, dz=3.88; *Figure 2A$_1$ and A$_2$*; see also *Figure 2—source data 1*). No significant correlation was found between emb-q-rating and proprioceptive drift.

In the control behavioral experiment, similar results were found for both proprioceptive drift and emb-q-rating (drift=mean ± sd: 2.47 ± 2.707 cm vs. 0.075 ± 2.461 cm; $t_{(19)}$=5.275, p=0.000043, dz=1.18; emb-q-rating=mean ± sd: 2 ± 0.763 vs. −0.97 ± 1.387; $t_{(19)}$=9.357, p=0.0000001, dz=-2.1; *Figure 2,B$_1$ and B$_2$*; see also *Figure 2—source data 2*). Furthermore, the disembodiment questionnaire rating (disemb-q-rating) was significantly higher in the synchronous than in the asynchronous condition (disemb-q-rating=mean ± sd: 0.153 ± 1.427 vs. −1.15 ± 1.371; $t_{(19)}$=3.835, p=0.00116, dz==0.86; *Figure 2,B$_3$*; see also *Figure 2—source data 2*). Finally, significant correlations were found between emb-q-rating and disemb-q-rating in both synchronous and asynchronous conditions and in the delta synchronous minus asynchronous (respectively: r=-0.4911, p=0.0279; r=-0.7128, p=0.0004; r=-0.5537, p=0.0113; *Figure 2,C$_1$,C$_2$,C$_3$*). No significant correlations was found between proprioceptive drift and either emb-q-rating or disemb-q-rating.

For the physiological data, the Friedman non-parametric ANOVA showed a significant effect of condition (χ2 [2, n=24]=9,000,000; p= 0.01111), suggesting a difference between baseline, synchronous and asynchronous conditions, when MEPs were recorded from the stimulated (right) hand. Wilcoxon matched pairs tests, after Bonferroni correction, revealed a significant MEP decrease in the synchronous condition with respect to both the asynchronous (mean ± sd: −0.367 ± 0.362 vs. 0.205 ± 0.395; Z=3.3143, p=0.000919; dz=0.85) and the baseline (mean ± sd: −0.367 ± 0.362 vs. 0.277 ± 0.691; Z=3 .1428; p=0.001673; dz=0.74) conditions (*Figure 3A$_1$*; see also *Figure 3—source data 1*). No significant difference was found between asynchronous and baseline conditions (0.205 ± 0.395 vs. 0.277 ± 0.691; Z=0.1714; p=0.863887; dz=0.08). Examples of MEPs recorded from the FDI muscle of a representative subject are shown in *Figure 3*. Interestingly, the MEP time-course analysis showed that the inhibitory motor response increases over time, with lower values measured at each time-point. Note that Wilcoxon matched pairs tests, after Bonferroni correction, showed a significant difference between TIME 1 (first five trials) and TIME 4 (last five trials) (mean ± sd: −0.23963 ± 0.425821 vs. −0.5481 ± 0.394248; Z=2.942857; p=0.003252; dz=0.72), *Figure 3A$_2$*; see also *Figure 3—source data 1*.

By contrast, in the control physiological experiment, in the three-level one-way ANOVA, no significant effect of condition was detected ($F_{(2,38)}$=0,894943; p=0.417068), suggesting that for the non-stimulated (left) hand there is no difference between the synchronous condition and either the asynchronous (mean ± sd: −0.104 ± 0.399 vs. 0.045 ± 0.262; p=0.820737; dz=0.26) or baseline (mean ± sd: −0.104 ± 0.399 vs. 0.0572 ± 0.368; p=0.711483; dz=0.22) conditions (*Figure 3B$_1$*; see also *Figure 3—source data 2*). Moreover, in the four-level one-way ANOVA adopted for analysis of MEP time-course, no significant effect of TIME was found ($F_{(3,57)}$=0,842673; p=0.476191), suggesting that in the non-stimulated (left) hand, MEP amplitude is not modulated over time (*Figure 3B$_2$*; see also

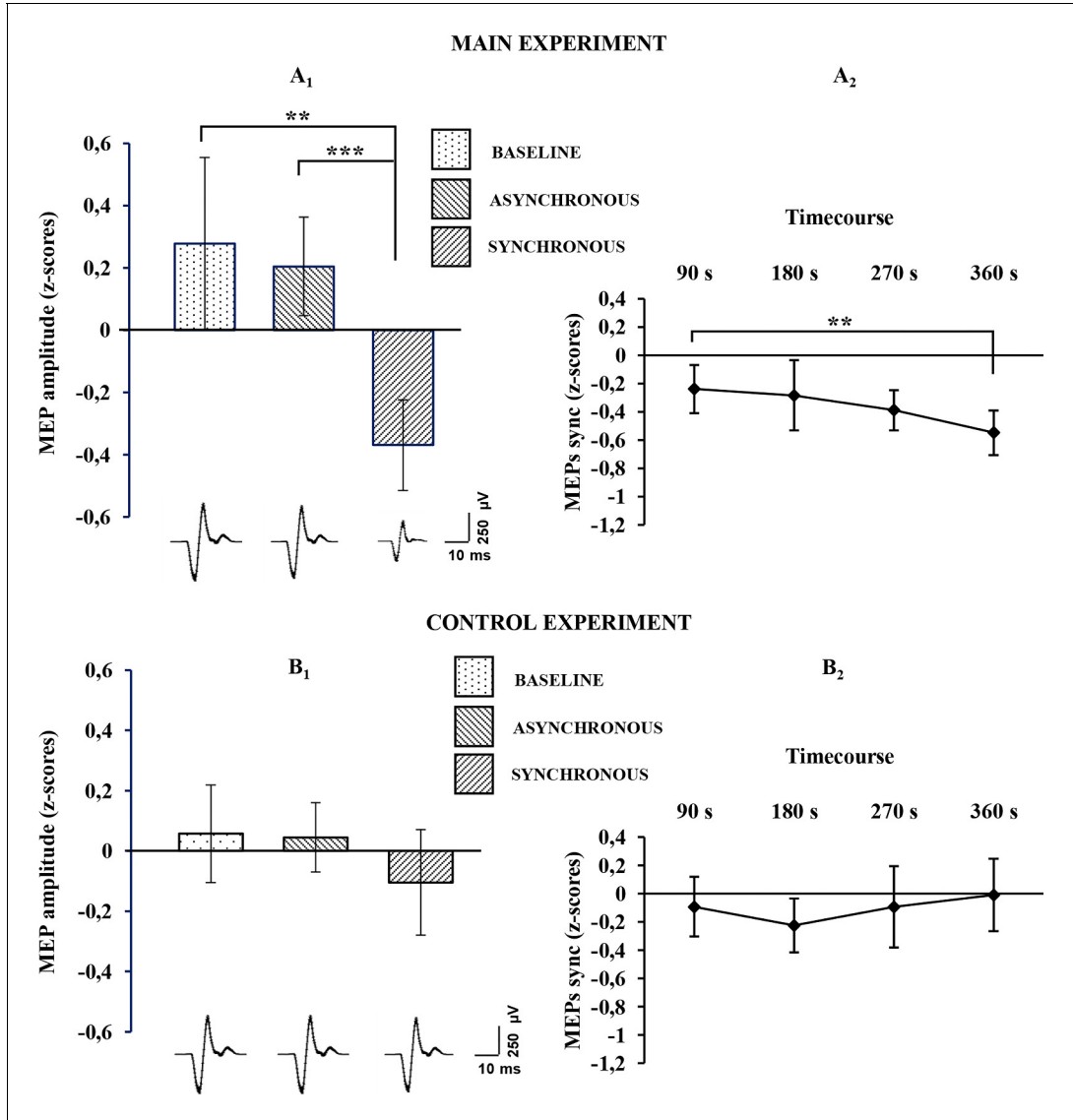

**Figure 3.** Physiological results for the baseline, asynchronous and synchronous conditions. Average MEP amplitude variation in the FDI muscle recorded across all subjects are plotted in $A_1$, for the main experiment, and in $B_1$ for the control experiments. Histograms represent the peak-to-peak MEP mean amplitude (normalized) ± 95% CI in the baseline, asynchronous and synchronous conditions, respectively. Significant levels: **$p<0.005$; ***$p<0.0001$. Average MEP amplitude profile recorded across all subjects in the synchronous condition areplotted in $A_2$ for the main experiment and in $B_2$ for the control experiment; points represent the peak-to-peak MEP mean amplitude (normalized), ± 95% CI, at four time-points after induction of the illusion (90 s, 180 s, 270 s 360 s); significance level: **$p<0.005$. Examples of average raw MEPs recorded from two representative subjects (for the main and control experiments) in the baseline (main: 609 µVolt; control: 619 µVolt), asynchronous (main: 771 µVolt; control: 601 µVolt) and synchronous (main:150 µVolt; control:583 µVolt) conditions. All subjects' physiological data are available in an additional source data file (see *Figure 3—source data 1* and *2*).

The following source data is available for figure 3:

**Source data 1.** Main experiment physiological results during baseline, asynchronous and synchronous condition.

**Source data 2.** Control experiment physiological results during baseline, asynchronous and synchronous condition.

*Figure 3—source data 2*). Additionally, the Mann-Whitney U-test revealed a significant difference between the delta synchronous minus asynchronous obtained in each experiment, showing a strong physiological effect of the RHI in the main compared to the control experiment (mean ± sd: −390.746 ± 591.662 vs. − 49.0296 ± 238.332; Z=2.074180; p=0.038063).

## Discussion

In the present study, in order to investigate the link between body-ownership and motor system, we took advantage of the well-established RHI paradigm (*Botvinick and Cohen, 1998*; *Ehrsson et al., 2004*; *Tsakiris et al., 2010*; *Longo et al., 2008*; *Moseley et al., 2008*; *Kammers et al., 2011*; *Rohde et al., 2013*; *Folegatti et al., 2009*; *Lewis and Lloyd, 2010*; *Valenzuela Moguillansky et al., 2013*; *Tsakiris and Haggard, 2005*), a useful tool to manipulate the sense of body ownership in normal subjects. In the main behavioral experiment, our data show a very strong embodiment effect in the synchronous condition (with a mean rating of 2.4, on a scale of −3 to +3), while none of the subjects reported the illusory experience in the asynchronous condition (with a mean rating of −2, on a scale of −3 to +3). A complementary disembodiment effect (*Longo et al., 2008*) was investigated and measured in the control behavioral experiment, showing a significant correlation between the reported ownership of the fake hand and disownership of the subject's own hand. Most importantly, the physiological results provide the first evidence that, during the RHI, the motor excitability of corticospinal hand circuits for the real stimulated hand is greatly reduced. This effect is absent for the real non-stimulated hand. In addition, consistent with behavioral studies reporting an increased illusory experience over time (*Lewis and Lloyd, 2010*; *Valenzuela Moguillansky et al., 2013*), the time-course analysis revealed that motor cortex excitability decreases as time of exposure to the illusion increases.

The link between body ownership and motor system activation has been investigated with different behavioral paradigms within the RHI framework. Recently, using a moving version of the RHI, it has been shown that voluntary (but not passive) movement of the real hand decreases the perceptual shift towards the rubber hand, suggesting that the subjective sense of agency strongly contributes to a coherent sense of body ownership (*Tsakiris et al., 2010*). On the other hand, patients with schizophrenia, who show a specific deficit in predicting the consequences of their voluntary actions (*Voss et al., 2010*), as well as an altered sense of agency (*Garbarini et al., 2016*; *Daprati et al., 1997*; *Maeda et al., 2012*), are more susceptible to the RHI (*Peled et al., 2003*; *Asai et al., 2011*; *Thakkar et al., 2011*). Other studies on clinical populations with movement disorders suggest that patients with focal hand dystonia (*Fiorio et al., 2011*) or partial or complete paralysis because of spinal cord injury (*Scandola et al., 2014*; *Tidoni et al., 2014*) seem to have some impairment of body ownership, according to their susceptibility to the RHI paradigm. A recent study (*Burin et al., 2015*), investigating the RHI in movement disorder after brain-damage, shows that hemiplegic patients display a weaker/more flexible sense of body ownership for the affected (paralyzed) hand (where the strength of the RHI is increased) when compared to controls, but an enhanced/more rigid sense of body ownership for the healthy hand (where the strength of the RHI is decreased). In other words, the prolonged absence of movement makes the paralyzed limb more readily disowned, while the healthy limb seems to be more strongly owned. Furthermore, other studies (*Kilteni et al., 2016*; *Schütz-Bosbach et al., 2006*) have investigated the modulation of corticospinal excitability during different experimental manipulations related to the sense of body ownership, though none with the specific hypothesis of linking a decrease in subjective ownership with a decrease in motor excitability. *Kilteni and colleagues (2016)* found that healthy people can experience a pseudo-amputation illusion during a virtual reality procedure, suggesting that this experimental manipulation causes corticospinal excitability changes in muscles associated with the virtually amputated body-part. *Schütz-Bosbach and colleagues (2006)* studied how an action observation task can induce different changes in corticospinal excitability, depending on the level of ownership experienced with respect to the observed moving hand. Ownership was previously manipulated using the RHI procedure: after asynchronous stimulation, observing others' actions facilitated the motor system, whereas after synchronous stimulation, identical observed actions, now illusorily attributed to the subject's own body, evoked smaller MEPs. In light of our findings, the absence of facilitatory effect during observation following synchronous stimulation can be interpreted as resulting from the decreased corticospinal excitability induced by the illusion.

Taken together, these results suggest that body ownership and the motor system are mutually interactive and both contribute to the dynamic construction of bodily self-awareness in healthy and pathological brains. In the present study, we found that the excitability of motor pathways in response to stimulation of the real (disembodied) hand is significantly decreased (i.e., MEP amplitude was significantly reduced) when subjects experience the artificial hand as their own. This suggests that an experimental manipulation of the sense of body ownership is accompanied by a coherent modulation of the motor system. However, as the experimental design was not suited to investigate correlations (see details in Material and methods), the present data do not allow us to address the question of whether subjects who experience a larger subjective illusion also show a larger decrease in MEPs. The presence or absence of linear correlations between MEP amplitude and behavioral measures, including both embodiment of the rubber hand and disembodiment of the real hand, will be investigated in future studies, in which physiological parameters of responder and non-responder subjects can be also compared.

It has been suggested that active movements integrate distinct body parts into a unitary body representation (*Tsakiris et al., 2010*). When this unified representation is altered, as during the visual-tactile conflict induced by the illusion, the excitability of the primary motor cortex is also altered, suggesting that the motor readiness of the real (disembodied) hand could also be reduced. We ascribe the excitability decrease in the primary motor cortex (M1) to cortical inhibitory processes tied to the central processing of hand ownership, possibly reaching M1 via inhibitory input from the premotor cortex (which is known to play a crucial role in the multisensory integration processes that give rise to the sense of body ownership) (*Ehrsson et al., 2004*).

The present findings, which shed new light on our understanding of the different aspects that contribute to the formation of a coherent self-awareness, suggest that bodily self-consciousness strictly depends on the possibility of movement. The bodily self is primarily and originally construed in terms of motor potentiality for actions (*Gallese and Sinigaglia, 2010*) . If I believe that the hand is mine, then I must be ready to use it; if not, then the activity of the motor system is accordingly down-regulated.

## Material and methods

### Participants

Twenty-six (ten male) volunteers took part in the behavioral main experiment (mean age ± SD=24 ± 5 years) (the sample size estimation was performed according to an *a priori* power analysis, see details in *Supplementary file 1*). Additionally, a different sample of 26 subjects (ten male) (mean age ± SD=24 ± 4 years) were recruited for the behavioural control experiment. Subjects participated in the physiological experiments only if, in the embodiment questionnaire, they gave a rating higher than zero in the synchronous condition. According to this criterion, 24 out of 26 subjects participated in the physiological main experiment; 20 out of 26 subjects participated in the physiological control experiment.

All participants were right-handed, as assessed with the Edinburgh Handedness Inventory, and were screened to exclude a family history of psychiatric, neurological or medical disease. The experimental protocol was approved by the Ethics Committee of the University of Milano and written informed consent was obtained from each subject in compliance with the rules of the 1964 Declaration of Helsinki.

### RHI experimental procedure

In both main (*Figure 1A*) and control (*Figure 1B*) experiments, the RHI was evoked by the synchronous stroking of the rubber hand and of the participant's own hidden hand (the location of stroking on the two hands was carefully matched) using the traditional visual-tactile stimulation (*Botvinick and Cohen, 1998*). Asynchronous stroking of a participant own hand and the rubber hand was utilized as a control condition, in which strokes were delivered spatially and temporally out of phase between the two hands. Participants sat with their forearms resting on a table, with their right hand inserted, palm down, in one of two identical compartments of a wooden box (59 cm × 33 cm × 15 cm); the rubber hand was placed in the left compartment, in egocentric position and aligned with the participant's shoulder. The upper lid of the box could be lifted or lowered to either

reveal or occlude the participant's view of the rubber hand in the left compartment, while his/her right hand was always out of view. The participant was able to see only the rubber hand being stroked by the experimenter's right hand, while the subject's right hand and the experimenter's left hand were always out of subject's view. The distance between the index finger of the rubber hand and the participant's own right index finger was 20 cm. A cloth was placed so as to hide both the participant's shoulder and the proximal end of the rubber hand. The behavioral RHI effect was measured in two ways: (1) by asking participants to localize the position of their unseen hand along the horizontal plan in front of them, and thus obtaining a measure of the proprioceptive drift towards the rubber hand, (2) with a questionnaire investigating their feeling of ownership of the rubber hand as a consequence of the experimental manipulation. Moreover, in the control experiment, we also measured the disembodiment experience of the stimulated hand with a questionnaire investigating the feeling of loss of own hand (*Longo et al., 2008*). The physiological effect was measured by recording motor-evoked potentials (MEPs), utilized to evaluate the excitability modulation of cortical and spinal motor neurons during the RHI. In the main experiment, MEPs were recorded from the real stimulated (right) hand, while in the control experiment, MEPs were recorded from the contralateral non-stimulated (left) hand.

## TMS and EMG recordings

Behavioral and physiological experiments were done in separate sessions: recording of MEPs in the three experimental conditions required about 40 min, and inserting the hand-ownership evaluation questions would have prolonged the experiment beyond a reasonable and feasible time, increasing the probability that subjects move their head with respect to the coil or lose their concentration on the task. In the main experiment, MEPs were elicited by single-pulse transcranial magnetic stimulation (TMS) of the hand area in the left M1 and recorded by self-adhesive bipolar surface electrodes that were placed over the belly of the right first dorsal interosseous muscle (FDI). Electromyographic (EMG) signals were amplified, filtered (10 Hz to 1 kHz), digitally converted (sampling rate 5 kHz) and stored in a computer for offline analysis. The head of each subject was restrained by a comfortable pillow wrapping around the neck and supported by a fixed head rest. A mechanical arm held a figure-of-eight-shaped coil connected to a magnetic stimulator (Magstim 200; Magstim Co. Ltd, Whitland, UK; maximal power 2.2 T). The coil was positioned and fixed on the left primary motor cortex with the handle pointing backwards at 45% from the midline so as to activate the selected muscle, and the stimulator output was set at about 110% of each subject's motor threshold (defined as the intensity giving three MEP responses out of six stimuli) (*Rossini et al., 1994*; *Borroni et al., 2008*). The absence of voluntary contraction before each TMS pulse was verified by continuous monitoring of the EMG signal.

We replicated the same procedure in the control experiment, but in this case, MEPs were elicited by single-pulse TMS of the hand area in the right M1 and recorded with self-adhesive bipolar surface electrodes over the left FDI belly.

## Experimental events sequence

### Baseline condition

At the very beginning of each experiment, after they gave their informed consent, subjects were asked to watch a cross on a pc screen while sitting in a comfortable chair. In the meantime, 20 MEPs were recorded (without visual-tactile stimulation) and taken as a measure of motor-cortex excitability in a neutral condition.

### Behavioral measurements

In both main and control experiments, participants were asked to judge the perceived location of their unseen right index finger by verbally indicating a number on a ruler presented on top of the box in front of them. This was repeated for 10 trials, and in each trial the ruler was shifted horizontally so as to avoid the subjects' basing their answers on a fixed reference point, rather than on their actual proprioceptive judgment. The difference between the indicated location of the participant's right index finger before and after the visual-tactile stimulation was taken as a measure of perceptual relocation.

In order to evaluate the subjective experience of the RHI, an ownership questionnaire (emb-q-rating) consisting of three statements was administered; participants were asked to evaluate the vividness of their experience of ownership of the rubber hand using a 7 points Likert scale (–3=strong disagreement; +3=strong agreement; 0=neither agreement nor disagreement), with the following three items: "*It seemed as if I were feeling the touch in the location where I saw the rubber hand touched*", "*It seemed as though the touch I felt was caused by the touch over the rubber hand*", "*I felt as if the rubber hand were my hand*". The statements were based on the traditional RHI study (*Botvinick and Cohen, 1998*). This behavioral part was performed both to replicate results found in previous studies (*Botvinick and Cohen, 1998*; *Ehrsson et al., 2004*; *Tsakiris et al., 2010*; *Longo et al., 2008*; *Moseley et al., 2008*; *Kammers et al., 2011*; *Rohde et al., 2013*; *Folegatti et al., 2009*; *Lewis and Lloyd, 2010*; *Tsakiris and Haggard, 2005*) on the behavioral RH effect and in order to include in the study only those subjects who experienced the illusion (with ratings higher than zero in the synchronous condition). Note that, according to this criterion, two subjects in the main experiment were not admitted to the physiological experiment. Moreover, in the control experiment, in order to verify whether the feeling of ownership of the RH is coherently accompanied by a feeling of disownership of the stimulated hand, a disownership questionnaire (disemb-q-rating) consisting of three statements was also administered. Participants were asked to evaluate the strength of their disembodiment experience over the stimulated hand using a 7-point Likert scale (–3=strong disagreement; +3=strong agreement; 0=neither agreement nor disagreement), with the following three items: "*It seemed like I was unable to move my hand*", "*It seemed like I couldn't really tell where my hand was*", "*I seemed like my hand had disappeared*". The statements were selected from a study proposing a psychometric approach to body ownership (*Longo et al., 2008*).

## Physiological measurements

For both main and control experiments, at the end of the behavioral procedure, 20 MEPs were recorded during both synchronous and asynchronous conditions. The order of the experimental blocks (synchronous-asynchronous; asynchronous-synchronous) was counterbalanced between subjects. Participants received an experimental block of 20 visual-tactile stimulations, either synchronous or asynchronous, depending on the random order of the sequence. Each visual-tactile stimulation cycle lasted 12 s. Two seconds after the end of each cycle, a single-pulse TMS was triggered to induce a MEP, using a custom-made synchronizing program in LabView10. After an inter-trial interval of 3 s, the next stimulation cycle started. Therefore, for each subject, 60 MEPs were recorded: 20 during the baseline conditions (at the beginning of the whole procedure) and 40 during the visual-tactile stimulation (20 during the synchronous block; 20 during the asynchronous). Participants were exposed to both experimental blocks (synchronous/asynchronous) in the same experimental session, with a resting break between the blocks (lasting about 5 min). The return of cortical excitability to baseline level was always ascertained before starting each block of visuo-tactile stimulation. The whole experimental procedure, including behavioral and physiological experiment, lasted about 40 min.

## Behavioral analysis

In both main and control experiments, the mean value of the three ownership statements used in the subjective rating questionnaire, in the synchronous and asynchronous conditions, was obtained and used as a dependent variable (emb-q-rating); in the control group, we also obtained and used as a dependent variable the mean value of the three disownership statements, in the synchronous and asynchronous conditions (emb-q-rating and disemb-q-rating). For the proprioceptive drift, the difference between the indicated location of the participant's right index finger before and after the visual-tactile stimulation (in both synchronous and asynchronous conditions) was taken as a measure of perceptual relocation, which was averaged and used as a dependent variable. All data were assessed for normal distribution using the Shapiro-Wilk test (p>0.05). In the main experiment, for the emb-q-rating in both synchronous and asynchronous conditions, the residuals were not normally distributed (p=0.00146 and p=0.0092), so the Wilcoxon signed-rank test was used for pairwise comparisons. In the control experiment, in both synchronous and asynchronous conditions for the emb-q-rating (p=0.08448 and p=0.60427) and the disemb-q-rating (p=0.88009 and p=0.34168), the

residuals were normally distributed, so comparisons between synchronous and asynchronous stimulations were computed by means of a paired T-test (two tailed). In both experiments, residuals for the proprioceptive drift were normally distributed (main: p=0.91975 and p=0.71247; control: p=0.90201 and p=0.37482), so comparisons between synchronous and asynchronous stimulation were computed by means of a paired T-test (two tailed). For each test performed, we reported mean, standard deviation, p (significance) value and Cohen's d value (calculated as within-subjects effect sizes using G Power matched pairs statistical tests). All subjects' behavioural data are available in an additional source data file (see *Figure 2—source data 1* and *2*).

## Physiological measures and analysis

MEP amplitude of the FDI muscle was measured as the peak-to-peak distance (in µV), and MEPs of amplitude lower than 50 µV were discarded from analysis (*Rossini et al., 1994*; *Borroni et al., 2008*). For each subject, 20 measurements of MEP baseline values were acquired at the very beginning of the experiment in order to provide a reference value that could be used a) to verify that, during the on-line data acquisition, the cortical excitability was unchanged in the second experimental block of visual-tactile stimulation compared to the first one and b) to compare, during data analysis, the obtained MEP values in the experimental blocks (in order to discriminate between facilitation or inhibition effects). Normal distribution of the residuals was checked using the Shapiro-Wilk test (p>0.05), and the appropriate non-parametric tests were applied when one or more of the corresponding data sets failed to meet the criteria for normal distribution. In both experiments, for each subject, we used MEPs recorded in each experimental condition (baseline, asynchronous and synchronous trials, a total of 60 trials), to obtain a grand-mean and a grand-standard-deviation. Then, each single trial was transformed in z-scores, according to the following formula: $(x – grand$-$mean)/(grand$-$standard$-$deviation)$, where $x$ indicates a single trial value. The obtained values were averaged for each subject and entered into two separate (for the main and control experiments) three-level (baseline, asynchronous, synchronous) one-way ANOVAs. In this analysis, for the main experiment, the distribution of residuals in the synchronous condition was not normal (respectively: p=0.70962, p=0.08347, p=0.00604). Thus, we performed Friedman non-parametric ANOVA in order to detect significant differences across the three conditions (baseline, asynchronous, synchronous); therefore, Wilcoxon signed-rank tests were used for pairwise comparisons. Finally, for each pairwise comparison (N = 3), a Bonferroni correction was applied (a value/n of comparisons: 0.05/3 = 0.017). In the control experiment, residuals of the three conditions were normally distributed (respectively: p=0.60988, p=0.44773, p=0.66546) and the ANOVA normality assumption was satisfied. Furthermore, we investigated the time course of MEP change in the synchronous condition of both experiments in order to describe, for each subject, a MEP amplitude time-profile during the illusion. MEP z-scores for all participants were divided into four blocks of five MEPs each. The obtained values were averaged starting from 0 to 5 (TIME 1), from 6 to 10 (TIME 2), from 11 to 15 (TIME 3) and from 16 to 20 (TIME 4), and entered in a four-level one way ANOVA (TIME: one, two, three, four). Residuals for the main experiment were not normally distributed, so Wilcoxon signed-rank tests were used for within comparisons of the four-level time variable; for each pairwise comparison (N = 4), Bonferroni correction was applied (a value / n of comparisons: 0.05/4 = 0.0125). Finally, in order to compare MEP modulation in the two experiments, we avoided classical ANOVA because residuals in the main experiment were not normally distributed. So, we calculated a delta on raw MEP amplitude between synchronous minus asynchronous for all subjects in each experiment; then the obtained values were analyzed with Mann-Whitney U-test. For each statistical test, we reported mean, standard deviation, p value and Cohen's d (calculated as within-subjects effect sizes using G Power matched pairs statistical tests). All subjects' physiological data are available in additional source data file (see *Figure 3—source data 1* and *2*).

## Correlation analysis

In the main experiment, linear regressions were performed between emb-q rating and proprioceptive drift in both synchronous and asynchronous conditions and in the delta synchronous minus asynchronous. In these correlations, the distribution of residuals, checked with Shapiro-Wilk test, was not normal, so we adopted the Spearman rank-order correlation. In the control experiments, linear correlations were performed between emb-q rating and disemb-q-rating in both synchronous and

asynchronous conditions and in the delta synchronous minus asynchronous; moreover, linear regressions between proprioceptive drift and emb-q rating /disemb-q-rating in both synchronous and asynchronous conditions and in the delta synchronous minus asynchronous were performed. In these correlations, residuals, when checked with the Shapiro-Wilk test, were normally distributed.

We acknowledge that the present experimental design was not ideally suited to investigate correlations, due to the fact that behavioural and physiological data were acquired in two separate sessions. Indeed, we could not obtain the behavioural responses during the registration of each MEP recording (due to time constraints during MEP acquisition) and, therefore, we could not have a point-by-point matching between those data. Furthermore, only responder subjects were admitted to the physiological experiment, i.e. we use only subjects who gave ratings higher than zero in the synchronous condition in the embodiment questionnaire administered during the behavioural experiment. Thus, in the present sample, which only includes responder subjects, correlations between physiological parameters and the presence/absence of the illusion cannot be investigated. However, to investigate whether responder subjects who experience a larger subjective illusion also show a larger decrease in MEP amplitude, we computed correlations, using either ratings of the questionnaire or proprioceptive drift values. In both cases, we used two different approaches: a) we normalized MEP values, ratings and drift values by using z-scores to compute independent correlations for synchronous and asynchronous conditions: significant correlations with MEPs were not found, neither for questionnaire ratings nor for proprioceptive drift; b) for MEP values, ratings and drift values, we computed a delta (synchronous minus asynchronous) value and performed correlations on these values: again, no significant correlations with MEPs were found for questionnaire ratings or for proprioceptive drift.

## Additional information

### Funding

| Funder | Grant reference number | Author |
| --- | --- | --- |
| Ministero dell'Istruzione, dell'Università e della Ricerca | RBSI146V1D MIUR-SIR | Francesca Garbarini |

The funders had no role in study design, data collection and interpretation, or the decision to submit the work for publication.

### Author contributions

FdG, Conception and design, Acquisition of data, Analysis and interpretation of data, Drafting or revising the article; FG, Conception and design, Analysis and interpretation of data, Drafting or revising the article; GP, Conception and design, Acquisition of data, Drafting or revising the article; AL, Acquisition of data, Drafting or revising the article; AB, PB, Conception and design, Drafting or revising the article

### Author ORCIDs

Francesco della Gatta, http://orcid.org/0000-0003-3471-6595
Francesca Garbarini, http://orcid.org/0000-0003-1210-0175
Annamaria Berti, http://orcid.org/0000-0001-9972-5543

### Ethics

Human subjects: All participants gave informed consent. The experimental protocol was approved by the Ethics Committee of the University of Milano and written informed consent was obtained from each subject in compliance with the rules of the 1964 Declaration of Helsinki.

## Additional files

### Supplementary files
• Supplementary file 1. Supplementary Materials.

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
