## [Decision Letter]

Thank you for submitting your article "Decreased motor cortex excitability mirrors own hand disembodiment during the rubber hand illusion" for consideration by *eLife*. Your article has been reviewed by two peer reviewers, including Alessandro Farné, as well as Reviewing Editor Jody Culham who also served as Senior Editor. The reviewers have discussed the reviews with one another and the Reviewing Editor has drafted this decision.

Summary of findings and import:

The authors report an interesting finding on the experimental manipulation of body ownership caused by the rubber hand illusion, the subjective feeling of the 'loss' of the participants' real hand. They report that the illusion results in decreased corticospinal excitability as measured with motor-evoked potentials. The approach demonstrates a new neural correlate of the illusion that provides insight into its effects on the motor system. Specifically, the findings show that the disembodiment of one's own hand is reflected in the motor system (whereas earlier behavioral work has focused on aspects of embodiment rather than disembodiment). As such, the results suggest that we "lose control" of our own hands if we don't feel that they are fully "ours".

Summary of decision:

All reviewers agreed that the paper is likely to make an interesting, timely and insightful contribution to the literature by showing that the rubber hand illusion leads to reduced cortical excitability (motor-evoked potentials) of the real hand. Nevertheless, they raised substantive concerns that would require a new experiment to address. *eLife* typically tries to avoid requesting new experimental data (and in such cases, the usual outcome is a rejection). However, during post-review discussion, the consensus was that as the paper was sufficiently promising and that the new experiment was likely (though not guaranteed) to work and yield a manuscript that would be publishable in *eLife*. Thus we encourage you to collect the recommended data and resubmit the manuscript, in which case we will solicit input from the same reviewers. Normally *eLife* aims to receive revisions within two months but if you expect this to be problematic, you can discuss it with the editor. Alternatively, given the extent of the revisions required, you may decide instead to submit your manuscript to another journal, in which case we ask you to notify us and we hope the reviewers' comments are helpful.

Essential points:

The following changes are necessary for publication.

1) Clearer analysis of MEP-illusion correlations

All three reviewers noted that the MEP-behavioral correlations were problematic, perhaps because the request after the initial submission wasn't specific enough or the desired analysis didn't work.

It appears that the new analysis (Figure 5) simply dichotomized the presence/absence of the illusion (which was essentially driven by the difference between the synchronous/asynchronous conditions) and correlated this with the continuous MEP magnitude. However, this doesn't tell us any more than a simple comparison of MEPs between synch and asynch (Figure 3). If you take a continuous measure of illusion strength for each subject (difference in continuous Q ratings for synch – asynch or difference in proprioceptive drift for synch – asynch) and do a scatterplot against difference in MEPs for each subject, is there a correlation? This would address the question of whether subjects who show a larger effect of the illusion also show a larger decrease in MEPs. This analysis should examine continuous variables, difference scores between synch and asynch, and try it with all the dependent behavioral variables (questionnaire scores and proprioceptive drift).

If those analyses don't show significant correlations, that doesn't mean that the paper shouldn't be published, but it may be worth including the analysis as a supplemental figure or a brief text description and tempering the conclusions accordingly.

2) Two reviewers agreed that the manuscript needs to make clearer the distinction between embodiment and disembodiment in the reduced MEPs.

The Authors' main conclusion is "… that a significant drop of motor excitability in M1 hand circuits accompanies the disembodiment of the real hand during the RHI experience." (Abstract, last sentence). The behavioural assessment provided here though is about embodiment, not disembodiment, which was not measured. The conclusion is thus drawn somewhat indirectly from the counterpart of disembodiment. Subjects were selected according to their sensitivity to the RHI, so the result that "… all subjects reported an illusory experience in the synchronous and not in the asynchronous condition." (Results, first paragraph) is a direct outcome of the inclusion criteria (subsection “Behavioural measurements”, last paragraph).

The paper would be much stronger if measures of *disembodiment* were included. Others have developed questions that address this (esp. Longo et al., 2008, http://www.ncbi.nlm.nih.gov/pubmed/18262508). The ideal solution would be to run a replication/extension of the experiment that also included questions about disembodiment (and perhaps some control questions to control for demand characteristics). It may also be that there are clearer correlations between the feeling disembodiment (than embodiment) and the decrease in MEPs.

3) Other improvements in experimental design should be included.

Ideally, the design should be better suited (a) to examine correlations between MEPs and behavioral measures, (b) to show that the change in MEPs is specific to the disembodied hand by obtaining control MEPs from the other non-stimulated hand, and (c) to collect baseline MEPs while the participants are looking at the rubber hand setup prior to induction of the RHI (rather than while looking at a computer screen).

Points that should be carefully considered:

It is not entirely clear how the procedure can take 40 minutes, when adding the time taken by each trial plus the breaks, please clarify.

Though mainly a question of terminology, some shortcuts would be best avoided. CSE (from FDI) is likely more than merely, or not that readily transposable to, a readiness to move the hand (e.g., Introduction, last paragraph, but elsewhere in the manuscript).

Please clarify some of the methods: how return to baseline was ascertained (subsection “Physiological measurements”); if you collected multiple baseline please provide statistics; the choice for discarding MEP (subsection “Physiological measures and analysis”) should also be clarified, together with the criteria applied to measure Rest Motor threshold intensity; Were MEP values normalised before Z-transformation?

One referee reviewed an earlier version of this experiment (n=14) submitted in another journal. In that version, the results were more or less identical with this one (n=24). It was puzzling to see this increase in the sample size and also the reference of the authors that they used an a priori power analysis. Please clarify.

In the previous version of the manuscript (n=14) the order presentation of the synchronous/asynchronous blocks was randomized, rather than counterbalanced. Here the authors report that the order was counterbalanced. Can you please confirm that the participants who took part in this study represent a new sample?

The authors used a top-up procedure of inducing the RHI, in cycles of 12 sec. Is there an effect of the accumulation of visuo-tactile stimulation? E.g. MEPs profile change as the illusion becomes stronger.

The authors should discuss this paper that seems highly relevant: Schütz-Bosbach S1, Mancini B, Aglioti SM, Haggard P. (2006) Self and other in the human motor system. Curr Biol. 16(18)[18]:1830-4. They also used MEPs as DV following the RHI, albeit the focus was slightly difference. However, given the conclusions you draw in relation to motor representations, and the findings of that past study, it is imperative that you discuss/contrast the two findings.

The figures make it confusing as to whether or not the subject could see the experimenter's hands stroking the rubber and real hands. Figure 1 implies the subject could see both of the experimenter's hands (but usually they only see the one stroking the rubber hand but not the one stroking the real hand) while Figure 4 implies they only saw part of the hand stroking the rubber hand and none of the hand stroking the rubber hand. It's also confusing that the asynchronous case in Figure 4 looks like the real hand is not being stroked at all.

[Editors' note: further revisions were requested prior to acceptance, as described below.]

Thank you for resubmitting your work entitled "Decreased motor cortex excitability mirrors own hand disembodiment during the rubber hand illusion" for further consideration at *eLife*. Your revised article has been favorably evaluated by Jody Culham as the Senior editor.

The manuscript has been improved but there are some remaining issues that need to be addressed before acceptance, as outlined below:

1) Please add a very brief description of the failure to find correlations (as described on the second page of the reply to reviewers). This should be mentioned – not in the main text but in the details at the end (perhaps in the Methods where you discuss why the design was suboptimal for correlations) or as supplemental materials. It's not that it increases the impact of the article, but rather that there is growing awareness in psychology (and hopefully other fields of science) that researchers need to report and acknowledge analyses that didn't work and not just those that did.

2) Although the reply letter seems to have been carefully edited, the manuscript itself was not and contained typos and misspellings. Please proofread the draft carefully before resubmission.

---

## [Author Response]

*Essential points:*

*The following changes are necessary for publication.*

*1) Clearer analysis of MEP-illusion correlations*

*All three reviewers noted that the MEP-behavioral correlations were problematic, perhaps because the request after the initial submission wasn't specific enough or the desired analysis didn't work.*

*It appears that the new analysis (Figure 5) simply dichotomized the presence/absence of the illusion (which was essentially driven by the difference between the synchronous/asynchronous conditions) and correlated this with the continuous MEP magnitude. However, this doesn't tell us any more than a simple comparison of MEPs between synch and asynch (Figure 3). If you take a continuous measure of illusion strength for each subject (difference in continuous Q ratings for synch – asynch or difference in proprioceptive drift for synch – asynch) and do a scatterplot against difference in MEPs for each subject, is there a correlation? This would address the question of whether subjects who show a larger effect of the illusion also show a larger decrease in MEPs. This analysis should examine continuous variables, difference scores between synch and asynch, and try it with all the dependent behavioral variables (questionnaire scores and proprioceptive drift).*

*If those analyses don't show significant correlations, that doesn't mean that the paper shouldn't be published, but it may be worth including the analysis as a supplemental figure or a brief text description and tempering the conclusions accordingly.*

In a previous version of study, mentioned by one of the reviewers, we presented a linear correlation between subjective ratings, recorded during the behavioural experiment, and MEPs amplitude, recorded during the physiological experiment. Reasoning on this correlation and on the critical comments of our previous reviewers, we decided not to include linear correlations in our pre-submission proposal to *eLife*. As we also mentioned in our submission letter, we are strongly convinced that the present experimental design was not ideal to investigate linear correlations and, in the revised version, we acknowledge this point as a limitation of the study in both Methods and Discussion sections, adding the following sentences:

Methods:

“We acknowledge that the present experimental design was not ideal to investigate correlations, due to the fact that behavioural and physiological data were acquired in two separate sessions. […] Thus, in the present sample, only including responder subjects, correlations between physiological parameters and the presence/absence of the illusion cannot be investigated”.

Discussion

“However, as the experimental design was not suitable to investigate correlations (see details in Methods), the present data do not allow to address the question of whether subjects who experience a larger subjective illusion also show a larger decrease in MEPs. The presence/absence of linear correlations between MEP amplitude and behavioral measures, including both embodiment of the rubber hand and disembodiment of the real hand, will be investigated in future studies, which will compare physiological parameters of responder and non-responder subjects”.

However, we computed on the present data all the required correlations, using either ratings of the questionnaire or proprioceptive drift values. In both cases, we used the two suggested approaches:

A) We normalized MEP values, ratings and drift values in z-scores to compute independent correlations for synchronous and asynchronous conditions: significant correlation with MEPs was found neither for questionnaire ratings nor for proprioceptive drift.

B) For MEP values, ratings and drift values we computed a δ (synchronous minus asynchronous) and we performed correlations on the obtained δ values: significant correlation with MEPs was found neither for questionnaire ratings nor for proprioceptive drift.

For the reasons mentioned above, we did not include these linear correlations in this revised version, but we are willing to include them if, after this clarification, the Editor and the Reviewers still think that this would strengthen the impact value of our study.

*With respect to the probit regression, we agree with the comment that it does not add significantly new results with respect to the main analysis. The probit model suggests that the subjects’ physiological parameters can significantly predict the presence/absence of the illusory experience, as reported during the behavioural experiment. In other words, this is a confirmative analysis of our main results showing that, starting from MEP values, the model classifies most cases (around 80%) in the correct way, predicting if they correspond to a condition in which the subjects experienced the illusion (synchronous condition) or not (asynchronous condition). For this reason, according to the reviewers’ suggestion, we removed this analysis from the revised version.*

*2) Two reviewers agreed that the manuscript needs to make clearer the distinction between embodiment and disembodiment in the reduced MEPs.*

*The Authors' main conclusion is "… that a significant drop of motor excitability in M1 hand circuits accompanies the disembodiment of the real hand during the RHI experience." (Abstract, last sentence). The behavioural assessment provided here though is about embodiment, not disembodiment, which was not measured. The conclusion is thus drawn somewhat indirectly from the counterpart of disembodiment.*

We agree with this comment and, in the revised Discussion, we acknowledged that in our main behavioural experiment we took for granted the relationship between embodiment of the fake hand and disembodiment of the real hand. Thus, according to the reviewers’ suggestion we designed a control behavioural experiment to explicitly investigate this point.

*Subjects were selected according to their sensitivity to the RHI, so the result that "… all subjects reported an illusory experience in the synchronous and not in the asynchronous condition." (Results, first paragraph) is a direct outcome of the inclusion criteria (subsection “Behavioural measurements”, last paragraph).*

We reasoned on this comment and, in the revised version, we included more details about the adopted criterion for the subjects’ selection: only subjects who, at the embodiment questionnaire administered during the behavioural experiment, gave rating higher than zero in the synchronous condition, were considered responder subjects and were admitted to the physiological experiment. According to this criterion, we included 24 out of 26 subjects. See Methods (first paragraph). However, we noticed and reported that in our sample all subjects reported a strong illusory experience in the synchronous condition (with a mean rating of 2.4 out of 3) and they did not report illusory experience in asynchronous condition (none of the subjects gave ratings higher than zero in the asynchronous condition and the mean rating was -2 out of -3). See Discussion (first paragraph). We think that this is not just a direct outcome of the inclusion criterion.

Note that, within the other reasons reported above, this inclusion criterion is, in our view, another point against the possibility to investigate linear correlations between physiological parameters and the presence/absence of the illusory experience because all included subjects were responders. See Methods (subsection “Correlation analysis”).

Note also that similar criterion for subject selection (only subjects who, at the embodiment questionnaire administered during the behavioural experiment, gave rating higher than zero in the synchronous condition, were considered responder subjects and were admitted to the physiological experiment) was used for the control experiment. According to this criterion 20 out of 26 subjects were admitted to the physiological control experiment. See Methods (first paragraph).

*The paper would be much stronger if measures of disembodiment were included. Others have developed questions that address this (esp. Longo et al., 2008, http://www.ncbi.nlm.nih.gov/pubmed/18262508). The ideal solution would be to run a replication/extension of the experiment that also included questions about disembodiment (and perhaps some control questions to control for demand characteristics). It may also be that there are clearer correlations between the feeling disembodiment (than embodiment) and the decrease in MEPs.*

According to this suggestion, we recruited a new sample and we performed the required two control experiments, a behavioural control experiment and a physiological control experiment (see below). In the behavioural control experiment, as suggested, we administered the disembodiment questionnaire of Longo et al. (2008) we found a significant disembodiment effect in the synchronous vs. the asynchronous condition and a significant correlation between the embodiment of the fake hand and the disembodiment of the real (disembodied) hand. See the revised Introduction (second paragraph), Methods (subsection “RHI experimental procedure”; subsection “Behavioural measurements”, last paragraph), Results (second paragraph) and Discussion (first paragraph).

*3) Other improvements in experimental design should be included.*

*Ideally, the design should be better suited (a) to examine correlations between MEPs and behavioral measures, (b) to show that the change in MEPs is specific to the disembodied hand by obtaining control MEPs from the other non-stimulated hand, and (c) to collect baseline MEPs while the participants are looking at the rubber hand setup prior to induction of the RHI (rather than while looking at a computer screen).*

We agree with the first two points:

A) In the revised version we acknowledged, as a limitation of the study, that the present design should be better suited to examine correlation; Methods (subsection “Correlation analysis, last paragraph), Discussion (third paragraph).

B) In the revised version, we recruited a new sample to performed the required physiological control experiment by recording MEPs from the non-deluded hand and we demonstrated that the change in MEP amplitude is specific to the deluded hand, see the revised Introduction (last paragraph), Methods (subsection “TMS and EMG recordings”; subsection “Physiological measures and analysis”), Results (third paragraph) and Discussion (first paragraph).

With respect to the last point (C), we think that recording MEPs while participants are looking at the rubber hand could be another experimental condition, because other studies have shown that only looking at the RH can modulate the subjects’ sense of body ownership, and also affect proprioceptive drift (see references below). However, in our design we focused on the comparison between synchronous and asynchronous condition, while the effect of looking at the RH was controlled, because it was equally present in both conditions. Thus, although the present data cannot exclude that the view of the RH per se can induce a MEP modulation with respect to the baseline, we can exclude that the inhibitory effect we found in the synchronous condition is merely due to observation of the RH during the experiment.

Farné, A., Pavani, F., Meneghello, F., & Ladavas, E. (2000). Left tactile extinction following visual stimulation of a rubber hand. Brain, 123, 2350–2360.

Pavani, F., Spence, C., & Driver, J. (2000). Visual capture of touch: Out-of-the-body experiences with rubber gloves. Psychological Science, 11, 353–359.

Durgin F, Evans L, Dunphy N, Klostermann S, Simmons K (2007) Rubber hands feel the touch of light. Psychol Sci 18: 152–157.

Rohde M, Luca M, Ernst MO (2011) The rubber hand illusion: Feeling of ownership and proprioceptive drift Do not go hand in hand. PLoS One 6(6)[6]:e21659. doi:10.1371/journal.pone.0021659.

*Points that should be carefully considered:*

*It is not entirely clear how the procedure can take 40 minutes, when adding the time taken by each trial plus the breaks, please clarify.*

The statement is now clarified it in the text, specifying that the entire experimental procedure, including behavioural and physiological experiment lasted about 40 minutes (plus the time for the correct localization of the FDI hot spot in M1, which was variable in each subject, and threshold determination). (Methods, end of subsection “Physiological measurements”).

*Though mainly a question of terminology, some shortcuts would be best avoided. CSE (from FDI) is likely more than merely, or not that readily transposable to, a readiness to move the hand (e.g., Introduction, last paragraph, but elsewhere in the manuscript).*

In order to avoid the shortcut, we now refer to “readiness of motor pathways”, a more literal description of a situation in which, with a lower excitability of primary motor cortex, a stronger voluntary command is going to be necessary to bring enough motorneurons to threshold and evoke movement.

*Please clarify some of the methods: how return to baseline was ascertained (subsection “Physiological measurements”); if you collected multiple baseline please provide statistics;*

We did not formally collect multiple baselines. Return to baseline was simply ascertained online by evoking a few MEPs (about 5 on average), without changing intensity of TMS stimulation, before starting each experimental session. It was important to do this between the second and third series of recordings, when subjects had undergone either the asynchronous or, especially, the synchronous stimulation, which could have long-term effects on MEP amplitude (although having the subject take a 5 min resting pause resulted sufficient to prevent this possibility). MEP amplitude is intrinsically rather variable and a precise measure would have indeed required recording a full 20-MEP series and comparing averages and variability; we decided that inserting an additional 20-MEP series would have excessively prolonged the overall experiment at the expenses of data collection in the experimental conditions of interest. Having worked with TMS on M1 for over a decade, we confided in our solid experience in evaluating the quality of the recorded signal.

*The choice for discarding MEP (subsection “Physiological measures and analysis”) should also be clarified, together with the criteria applied to measure Rest Motor threshold intensity;*

Resting motor threshold (MT) is defined as the minimal intensity required to elicit, with 50% probability, an EMG response of at least 50µV, in a fully relaxed muscle (Rossini et al., 1994), where the 50% probability is often measured as 5 MEPs elicited in 10 consecutive pulses (Rossini et al. 1994), but different numbers have been used, depending on the goal of the study. 3 MEPs in 6 pulses is also common practice (e.g. Borroni et al., 2008). MEPs smaller than 50µV are not considered true motor responses, either because they can be lost in noisy recordings or because at both extremes (very small or very large) MEP amplitude loses it linear relation with stimulus intensity.

Rossini PM et al. Non-invasive electrical and magnetic stimulation of the brain, spinal cord and roots: basic principles and procedures for routine clinical application. Report of an IFCN committee. Electroencephalogr. Clin. Neurophysiol. 1994; 91:79–92.

Borroni P, Montagna M, Cerri G, Baldissera F. Bilateral motor resonance evoked by observation of a one-hand movement: role of the primary motor cortex. European J Neurosci. 2008;28:1427–1435.

*Are MEP values normalised before Z-transformation?*

We used z-transformation to normalize our data, so no previous transformations were performed.

*One referee reviewed an earlier version of this experiment (n=14) submitted in another journal. In that version, the results were more or less identical with this one (n=24). It was puzzling to see this increase in the sample size and also the reference of the authors that they used an a priori power analysis. Please clarify.*

*In the previous version of the manuscript (n=14) the order presentation of the synchronous/asynchronous blocks was randomized, rather than counterbalanced. Here the authors report that the order was counterbalanced. Can you please confirm that the participants who took part in this study represent a new sample?*

In the mentioned version of our study, we performed a similar experiment on a sample of 14 subjects. We received constructive critics on that work in different contexts, including the review processing mentioned by the referee and other comments received during conference meeting presentations. According to these suggestions, we replicated the experiment in a larger sample and we estimated the sample size using an a priori power analysis.

With respect to the randomized/counterbalanced problem, if we have had the possibility to answer to the reviewer in the previous revision round, we would have simply acknowledged a mistake in our method description: in both versions of our study, we always used a counterbalanced order.

However, the main difference between our previous study on 14 subjects and the present study on 24 subjects is related to the baseline condition. In the previous study we used as dependent variable of the MEP analysis only the synchronous and the asynchronous conditions and we plotted the baseline value as a graphical indication on the modulation direction (inhibition) in the synchronous condition. In that sample, we acquired 5 baseline trials at the beginning of the experiment and we used the mean value to plot the baseline in the graph. However, in the last 8 subjects we also acquired a true baseline condition with 20 trials (as in the synchronous and asynchronous conditions). We did this because, while running the experiment, we realized that a statistical comparison with the baseline could be useful. Then, when we decided that we needed to increase the sample size, we always included a baseline condition with 20 trials (as in the last 8 subjects of our previous study). Thus, in the present sample we included those 8 subjects from the previous data set and 16 new subjects: all performed the same experiment, including the baseline condition with 20 trials. Note that, 18 new subject were recruited for the behavioural experiment (thus, adding the previous 8 subjects, we had a sample of 26 subjects) but 2 out of 18 subjects were not admitted to the physiological experiment because they did not experience the illusion in the synchronous condition (see the above mentioned inclusion criterion to be admitted to the physiological experiment).

*The authors used a top-up procedure of inducing the RHI, in cycles of 12 sec. Is there an effect of the accumulation of visuo-tactile stimulation? E.g. MEPs profile change as the illusion becomes stronger.*

We thank the reviewers very much for this comment, which allowed us to perform an additional time course analysis of MEP values (see Methods subsection “Physiological measures and analysis”). This analysis showed that, consistent with behavioural studies reporting an increased illusory experience over time (see references below), the inhibitory motor response significantly increases during exposure to the illusion (see Introduction, last paragraph) (see Results, third paragraph).

Lewis E, Lloyd DM (2010) Embodied experience: A first-person investigation of the rubber hand illusion. Phenomenology and the Cognitive Sciences, Vol 9(3)[3], 317-339; http://dx.doi.org/10.1007/s11097-010-9154-2

Valenzuela Moguillansky C, O’regan JK, Petitmengin C (2013). Exploring the subjective experience of the “rubber hand” illusion”. *Front Hum Neurosci;* 7:659 10.3389/fnhum.2013.00659

*The authors should discuss this paper that seems highly relevant: Schütz-Bosbach S1, Mancini B, Aglioti SM, Haggard P. (2006) Self and other in the human motor system. Curr Biol.16(18)[18]:1830-4. They also used MEPs as DV following the RHI, albeit the focus was slightly difference. However, given the conclusions you draw in relation to motor representations, and the findings of that past study, it is imperative that you discuss/contrast the two findings.*

We added in the Discussion 2 previous studies investigating the modulation of corticospinal excitability during different experimental manipulations related to the sense of body ownership. Together with the suggested Schütz-Bosbach et al. paper, we also mentioned the recently published Kilteni et al. paper. See revised Discussion (second paragraph).

*The figures make it confusing as to whether or not the subject could see the experimenter's hands stroking the rubber and real hands. Figure 1 implies the subject could see both of the experimenter's hands (but usually they only see the one stroking the rubber hand but not the one stroking the real hand) while Figure 4 implies they only saw part of the hand stroking the rubber hand and none of the hand stroking the rubber hand. It's also confusing that the asynchronous case in Figure 4 looks like the real hand is not being stroked at all.*

We thank the reviewers for these suggestions. As usually done in RHI works, also in our study subjects could only see the rubber hand and the right hand of the experimenter (the one which touches the RH). The participant’s right hand and the left experimenter’s hand were always hidden out from participant’s view. We modified small details in order to make it more intuitively clear (see Figure 1 and capture; see Methods subsection “RHI experimental procedure”).

[Editors' note: further revisions were requested prior to acceptance, as described below.]

*1) Please add a very brief description of the failure to find correlations (as described on the second page of the reply to reviewers). This should be mentioned – not in the main text but in the details at the end (perhaps in the Methods where you discuss why the design was suboptimal for correlations) or as supplemental materials. It's not that it increases the impact of the article, but rather that there is growing awareness in psychology (and hopefully other fields of science) that researchers need to report and acknowledge analyses that didn't work and not just those that did.*

We agree that reporting negative data is important and, according to the Editor’s suggestions, we reported a brief description of the failure to find correlations in methods, where we discussed why the design was suboptimal for correlations. See subsection “Correlation analysis”, last paragraph.

2) Although the reply letter seems to have been carefully edited, the manuscript itself was not and contained typos and misspellings. Please proofread the draft carefully before resubmission.

We apologize for typos and misspellings. We realized that in our last version of the manuscript the English spell-checker in word processing software was turned off. To edit the present version, we turned on the spell-checker and we carefully revised the English, also according to the Editor’s flagged file. We would like to thank very much the Editor for proofreading our manuscript and including the copy of her revision. It was very useful!